# Current Aspects of Selected Factors to Modulate Brain Health and Sports Performance in Athletes

**DOI:** 10.3390/nu16121842

**Published:** 2024-06-12

**Authors:** Katarzyna Przewłócka, Daria Korewo-Labelle, Paweł Berezka, Mateusz Jakub Karnia, Jan Jacek Kaczor

**Affiliations:** 1Division of Physiology, Gdansk University of Physical Education and Sport, 80-336 Gdansk, Poland; katarzyna.przewlocka@awf.gda.pl; 2Department of Physiology, Faculty of Medicine, Medical University of Gdansk, 80-211 Gdansk, Poland; daria.korewo-labelle@gumed.edu.pl; 3Department of Animal and Human Physiology, Faculty of Biology, University of Gdansk, 80-309 Gdansk, Poland; pawel.berezka@phdstud.ug.edu.pl (P.B.); mateusz.karnia@ug.edu.pl (M.J.K.)

**Keywords:** stress, supplements, sports performance, brain and muscle regeneration

## Abstract

This review offers a comprehensive evaluation of current aspects related to nutritional strategies, brain modulation, and muscle recovery, focusing on their applications and the underlying mechanisms of physiological adaptation for promoting a healthy brain, not only in athletes but also for recreationally active and inactive individuals. We propose that applying the rule, among others, of good sleep, regular exercise, and a properly balanced diet, defined as “SPARKS”, will have a beneficial effect on the function and regeneration processes of the gut–brain–muscle axis. However, adopting the formula, among others, of poor sleep, stress, overtraining, and dysbiosis, defined as “SMOULDER”, will have a detrimental impact on the function of this axis and consequently on human health as well as on athletes. Understanding these dynamics is crucial for optimizing brain health and cognitive function. This review highlights the significance of these factors for overall well-being, suggesting that adopting the “SPARKS” approach may benefit not only athletes but also older adults and individuals with health conditions.

## 1. Introduction

Post-exercise recovery is of the utmost importance for athletes. Effectively managing the equilibrium between training stress and recovery is essential to optimizing adaptation and performance during training as well as the competition period [1]. The physiological and psychological demands of a competitive season may present significant challenges to athletes. Consequently, athletes should balance stress and recovery, employing various techniques to effectively manage fatigue and improve overall recovery and performance in subsequent training sessions and competitions. It is widely recognized that sports performance is a comprehensive and multifaceted process that involves the musculoskeletal system as well as the central nervous system (CNS) and the peripheral nervous system (PNS) [2]. Changes in the CNS and motor unit recruitment are widely believed to be directly linked to fatigue, resulting in declines in both physical and mental performance [3]. Physical fatigue can be attributed to various factors, including muscle damage, glycogen depletion, dehydration, and mental fatigue [3]. Therefore, this is an appropriate time to summarize our current knowledge about nutritional strategies, brain modulation, and muscle recovery, highlighting their applications and the underlying mechanisms of physiological adaptation for fostering brain health in both athletes and non-athletes.

This review presents recent advancements in selected nutrition strategies, brain modulation, and muscle recovery, with a focus on promoting brain health in competitive athletes and individuals with different activity levels. A comprehensive bibliographic search was conducted in PubMed to identify pertinent articles and experimental evidence, offering readers a thorough analysis of the subject matter.

Athletes face significant stress due to the nature of their training, which includes both high-intensity intermittent activity and prolonged exercises, leading to considerable physiological and neuromuscular stress [4]. While stress is crucial for physiological adaptation and performance enhancement [5], chronic stress can worsen inflammation, disrupt homeostasis, and impair muscle function and sports performance. The skeletal muscle, abundant in the human body, not only facilitates movement but also regulates systemic metabolic balance and influences responses throughout the organism by releasing signaling factors [6]. Therefore, muscle contraction may affect many tissues and organs. Emerging evidence suggests that the CNS is also subject to signaling initiated by muscles [6], a topic we will delve into further in this review. Moreover, there is an assumption that a gut–brain–muscle axis exists and plays a significant role in maintaining not only physical but also mental health, which is crucial not only for athletes. Hence, proper nutrition and organism recovery strategies are essential to maintain the optimal function of this axis. By exploring the interplay between nutrition strategies, brain modulation techniques, and muscle recovery, valuable insights may be gained to promote overall brain health and cognitive function, thus enhancing sports performance.

This review offers a thorough evaluation of nutritional strategies, brain modulation, and muscle recovery, focusing on promoting a healthy brain in competitive athletes and individuals with varying activity levels. By exploring their applications and underlying physiological mechanisms, it sheds light on the importance of considering specific factors to foster a healthy brain and enhance cognitive function. Athletes are exposed to several negative external and internal factors that may disrupt sports performance and recovery processes. A cascade of negative events associated with chronic stress can initiate overtraining and sleep disorders, resulting in mood changes. Poorly balanced nutrition may lead to dietary neglect, causing dysbiosis, with increased inflammatory and oxidative stress that affects many tissues and organs, directly increasing the risk of injury or infection (SMOULDER). To prevent the negative effects of chronic stress, elements described by the acronym SPARKS should be routinely implemented as preventive measures. These measures encompass both psychological-cognitive aspects (quantity and quality of sleep and post-workout recovery) and ensuring proper bodily functions, including a balanced diet rich in probiotics, antioxidants, and supplements to improve the physiological function of not only athletes. We advocate for adhering to the “SPARKS” rule for beneficial outcomes, contrasting it with the detrimental effects of embracing the “SMOULDER” formula (see Figure 1).

## 2. Brain Modulation

### 2.1. The Interplay between Selected Internal and External Factors and Sports Performance

#### 2.1.1. Epidemiology of Stress

As shown by epidemiological data, in 2018, 74% of British people felt so stressed that they had been overwhelmed or unable to cope. Moreover, the same data have shown that 51% of adults who felt stressed reported feeling depressed, and 61% reported feeling anxious [7]. Also, the most recent data covering COVID-19 indicate a significant percentage of the population to be suffering from chronic stress. The results of extensive reviews show that the prevalence of people experiencing stress in the general population and healthcare workers is 30 and 43%, respectively. Furthermore, the prevalence of anxiety is similarly estimated as 32 to 37%, and the prevalence of depression as 34 to 35 [8,9].

#### 2.1.2. Stress Prolegomena

It is commonly assumed that due to a neurohormonal reaction to a stress stimulus, adrenal hormones are released into the general circulation: catecholamines, adrenaline, and noradrenaline from the adrenal medulla and glucocorticoids (GCs) from the adrenal cortex [10]. This viewpoint has been accepted since it was recognized that understanding the systemic effects of these hormones requires considering their role in the body’s adaptive response to environmental stress.

Selye coined the term General Adaptation Syndrome (GAS) to describe a systemic response to various internal and external stressors, encompassing both positive and negative stimuli [11]. GAS unfolds in three discernible phases: alarm, resistance, and exhaustion. If the originating stressors persist unchecked, they can have deleterious effects on physical and mental health. The primary mechanism of this reaction is to be realized through the so-called pituitary–adrenal axis that constitutes a multi-link chain of neurohormonal interaction. The adrenocorticotropic hormone (ACTH) released in the anterior pituitary gland, stimulating the adrenal cortex, leads to the seeding of cortical steroid hormones (mainly glucocorticoids—GCs), which triggers the appropriate peripheral adaptive response—stress.

While stress is not a mental health problem in and of itself, experiencing overwhelming stress for an extended period is often called chronic or long-term stress, and it can impact both physical and mental health [8].

#### 2.1.3. Detrimental Effects of Chronic Activation of the Hypothalamic–Pituitary–Adrenal Axis

The stressor load instigates a neuroendocrine cascade known as the hypothalamic–pituitary–adrenal (HPA) axis. This process is initiated in the hypothalamus, where corticotropin-releasing hormone (CRH) is secreted, leading to the stimulation of the anterior pituitary gland and the release of ACTH. The final result of this axis is the activation of the adrenal cortex, prompting the synthesis and secretion of GCs. Therefore, under environmental stress or pathophysiological conditions, such as starvation, coldness, or cancer, the circulating GCs levels are significantly increased, decreasing the rate of protein synthesis and raising proteolysis to generate amino acids to serve as precursors for hepatic gluconeogenesis. Thus, the destructive role of GCs is well established, and the catabolic action of GCs affects the brain [12], bone [13], liver, heart [14], and skeletal muscles [15]. The indicated alterations can exert a direct influence on athletes, resulting in an augmented susceptibility to musculoskeletal injuries [16], an increased prevalence of respiratory infections [17], heightened nociceptive responsiveness [18], and lastly, substantial disturbances to sleep patterns [19]. The emotional and cognitive reactions to stress that occur during a chronically persistent stress response may take many forms, but among the most common are emotional reactions that involve negative affect. Furthermore, cognitive and emotional responses to stress are considered among the main factors contributing to sleep disorders [20], and as research shows, many athletes suffer from poor sleep quality, disrupting the proper stress response [21].

#### 2.1.4. Sleep

The quality and quantity of sleep are crucial for maintaining well-being and brain and muscle functions. There is a hypothesis that during the sleep period, the brain can “turn off”, creating the proper conditions for neuronal connection regeneration [22]. It is established that sleep enhances neurometabolic, somatic, and cognitive functions. The somatic aspect of sleep plays a vital role in enhancing neurometabolic, somatic, and cognitive functions. The somatic aspect of sleep involves tissue restoration and support for the immune and endocrine systems, while cognitively, it positively impacts learning, memory processes, and synaptic plasticity [23]. An adequate sleep amount (seven or more hours per night) highly determines sports performance [24]. Thus, nutritional strategies targeted at improving sleep and facilitating regeneration are essential for athletes.

Sleep, regulated by the sleep–wake cycle and circadian rhythm, is overseen by the suprachiasmatic nucleus (SCN) in the brain. The SCN prompts melatonin secretion as darkness falls, promoting sleep [25]. Physiological changes during sleep occur in two primary states: non-rapid eye movement (NREM) and rapid eye movement (REM) sleep [26]. REM is characterized by high brain activity, which seems crucial, especially for motor skills and muscle regeneration [27]. During the REM period, corticospinal pathways are partially blocked by brainstem mechanisms, accounting for the brain–body disconnection. As a result, spinal motor neuron activity is inhibited, causing total muscle relaxation and creating conditions for myofibril restoration [28].

On the contrary, during NREM, low cortical activity is observed, as manifested in slow wave activity on electroencephalography (EEG) images [29]. This phase is associated with neuronal network recruitment and recovery at various levels, including neuronal excitability regulation, energy storage replenishment, synaptic plasticity, cellular membrane regeneration, and cellular homeostasis maintenance [30]. Undisturbed sleep cycles between the REM and NREM stages continuously, ensuring all the sleep functions occur [28]. Thus, sleep is recognized as a vital recovery strategy for athletes.

Post-exercise recovery is crucial for all athletes to avoid exhaustion and temporary impairments in physiological adaptation and sports results [31]. Myofibril damage induced by exercise triggers the adaptation process. However, chronic overtraining and inadequate recovery can lead to prolonged inflammation, impairing sports capabilities [32]. Pro-inflammatory cytokine overproduction, reactive oxygen species (ROS) generation, muscle soreness, and fatigue may additionally adversely affect sleep, contributing to mood deterioration, training motivation reduction, impairment of the autonomic nervous system, the elevation of catabolic hormones level, and thus, a decrease in anabolic processes. As a result, muscle protein synthesis deterioration and blunted sports performance are observed [33,34].

Conversely, sleep deprivation may negatively affect sports performance due to leading to endocrine and immune system impairment. As a result, reduced immunity and increased vulnerability to infections were observed [35]. Moreover, negative changes may be related to a shift in hormone secretion, pro-inflammatory cytokines, and C-reactive protein (CRP) release [36]. During the sleep period, the release of hormones involved in protein synthesis occurs [37]. Anabolic hormones like testosterone, growth hormone, and insulin growth factor 1 (IGF-1) strongly influence the skeletal muscles via activation of the phosphatidylinositol-3 kinase/protein kinase B pathway (PI3K/Akt) [37]. In addition, continuous awaking triggers cortisol secretion, promoting catabolism. Leproult and coworkers have demonstrated that one day of sleep disturbances was enough to cause a cortisol level elevation [38]. Thus, the sleep period is crucial for the regeneration process, contributing to tissue repair and neuronal system refresh.

Moreover, it was established that the sleep phase is a critical link between physiological adaptations to training, cognitive processes, tissue repair, and metabolic functions [39,40]. It could be distinguished into the main components: total time of sleep, quality as well as homeostasis of sleep; disruption to any of these may impair sports performance [35]. It has been noticed that sleep problems are commonly known in athletes. The quality and quantity of sleep may be disrupted by pre-competition anxiety, jet lag, lengthy travels, or training schedules [41,42]. Killer et al. observed that athletes who underwent short-term intensified training spent more time in bed without an increase in the effective sleep time. In addition, the number of awakenings per night and the movement time increased, indicating an elevated sleep fragmentation index [43].

Some studies have shown that, after sleep loss, the procedural memory, motor skills, and regeneration may be affected. Abedelmalek et al. found that one night of partial sleep deprivation correlated with an elevated plasma interleukin-6 (IL-6) concentration after short-term maximal exercises during recovery [44]. In another study, the same researchers showed that the concentration of IL-6 and tumor necrosis factor (TNF-α) was significantly higher during exercise and remained higher during the 60 min recovery period in athletes who underwent sleep shortages [45]. Other clinical studies have highlighted the role of sleep in the anaerobic capacity. Souissi et al. conducted a study on judokas, showing evidence that 4 h of sleep deprivation correlates with decreased muscle strength, peak power, and mean power during an afternoon Wingate test session compared with a night of standard sleep [46]. Both total and partial sleep deprivation may also be related to decreased aerobic capacity and muscle strength. In the study by Cullen et al., the aerobic capacity, maximal handgrip strength, and countermovement jump were reduced after total sleep deprivation by 11, 6, and 11%, respectively. At the same time, the aerobic capacity and countermovement jump were impaired by 4 and 5% following partial sleep disturbance among recreationally active males [41].

An inadequate amount and quality of sleep may also cause a decline in mental functions and well-being [47]. Therefore, increasing the sleep duration is associated with benefits in terms of alertness, psychomotor functions, executive function performance, and motivation level [47]. To sum up, the data mentioned above strongly support the importance of the sleep amount (seven or more hours) and quality as a critical factor promoting optimal regeneration processes, benefiting both the central nervous system and skeletal muscle tissue health.

## 3. Nutritional Strategies

### 3.1. Supplementation Supporting Performance/Recovery from Exercise-Induced Muscle Damage

#### 3.1.1. Melatonin

Melatonin (*N*-acetyl-5-methoxytryptamine) is primarily produced by the pineal gland, with smaller amounts synthesized by other organs like the gastrointestinal tract, liver, and adrenal cortex [48]. Its rhythmic secretion is regulated by the circadian cycle, influencing sleep timing, nocturnal blood pressure, and body temperature [49]. Moreover, melatonin is a highly effective molecule protecting against oxidative stress, regulating energy metabolism, and affecting the immune system [50,51,52]. However, extra-pineal gland melatonin production is not dependent on the circadian rhythm. Scientific evidence has confirmed the common absence of melatonin fluctuation changes dependent on the night/day ratio in most extra-pineal tissues [53]. In addition, many tissues demonstrate higher intracellular melatonin concentrations than plasma, and the hormone is not released into the blood circulation. These data suggest that extra-pineal melatonin may locally protect cells against oxidative stress and inflammation. Thus, melatonin has been established as a highly effective antioxidant [53,54].

Tryptophan (Trp) is crucial in brain melatonin synthesis as the primary precursor. Trp must pass through the brain–blood barrier (BBB), which competes with other neutral amino acids, thus creating higher plasma tryptophan concentrations favorable to transportation into the brain [55]. The melatonin synthesis occurs in a four-step process. Trp is previously converted by tryptophan hydroxylase followed by aromatic amino acid decarboxylase, creating 5-hydroxytryptamine (5-HT) or serotonin [56]. 5-HT to melatonin conversion takes place in the pineal gland and the periphery via the successive action of two enzymes: serotonin *N*-acetyltransferase and hydroxyindole-O-methyltransferase. The serotonin pathway is established to exist in the brain, pineal gland, and gastrointestinal tract [57]. Trp-rich food like milk, cheese, poultry, fish, eggs, pumpkin seeds, nuts, beans, and leafy green vegetables may enhance melatonin synthesis and improve sleep quality [35]. Some studies indicate that consuming Trp-rich food in the evening may improve sleep quality and augment sustained alertness early in the morning [58,59].

Melatonin is a vital output signal from the SCN that generates the circadian rhythm and affects sleep. Its synthesis is stimulated by nocturnal norepinephrine released from pinealocytes through a sympathetic pathway [60]. The highest melatonin secretion occurs 2 h before an individual’s habitual bedtime and increases with the onset of sleepiness. The endogenous secretion of melatonin is escalated during the night, and the strong relationship between the timing of human sleep and the melatonin level has underlined the importance of this hormone in sleep regulation [61]. Thus, exogenous melatonin intake seems to enhance sleep quality, which Gorfine et al. have documented by neuroimaging in the healthy adult population [62]. The primary mechanism through which the neurohormone melatonin regulates circadian rhythm and sleep is by activating melatonin receptors (MT1 and MT2 in humans) in the hypothalamus, which belong to the family of G-protein coupled receptors. This results in the stimulation of G-proteins, as well as inhibition of adenylyl cyclase, and a decrease in the intracellular cyclic adenosine 3′,5′ monophosphate (cAMP) concentration occurs [63]. The NCS receives signals from the retinohypothalamic tract depending on the light availability and accordingly controls melatonin synthesis by the pineal glands, which alter the NCS activity via the MT1 and MT2 receptors’ feedback mechanism [64]. The physiological role of these receptors is not fully discovered. However, they may have complementary or opposite functions at both the central and peripheral levels. MT2 receptors are situated in the reticular thalamus (RT) and are mainly engaged in NREM sleep regulation.

Moreover, they are involved in the control of the circadian rhythms, dopamine release in the retina, and vasodilatation. Conversely, MT1 receptors located in the basal forebrain mainly affect REM sleep and are related to metabolic and reproductive functions and vasoconstriction [48,63]. Because of these receptors’ complementary and opposite roles, exogenous melatonin intake does not significantly affect REM and NREM sleep.

In addition, melatonin peaks are observed between 1 and 3 p.m. in nocturnal and diurnal species [65]. The above-mentioned data lead us to hypothesize that melatonin may improve sleep through acting as a pacemaker influencing the circadian rhythms rather than a soporific neuromodulator. Melatonin presumably enhances sleep by regulating MT1 and MT2 receptor expression during the peak [63].

An excessive training load may affect the circadian clock and disturb melatonin production via the elevation of TNF-α. It was shown that this pleiotropic cytokine suppresses Period gene (PER) expression, contributing to a prolonged rest time at dark. This process was described as promoting fatigue development [66]. Moreover, TNF-α and melatonin demonstrate bidirectional interaction [67]. As melatonin is a molecule displaying antioxidant and anti-inflammatory properties, TNF-α can inhibit transcription of the AA-NAT gene, which encodes alkylamine *N*-acetyltransferase, the key enzyme in melatonin biosynthesis and, thus, circadian rhythm regulation. However, this inhibition seems to be transient [68]. Plenty of studies indicate the beneficial role of exogenous melatonin in sports performance and well-being among athletes. The positive effect is associated with brain and skeletal muscle regeneration via sleep quality and duration improvement, together with melatonin’s anti-inflammatory and antioxidant properties. In the study by Leonardo-Mendonca et al., four weeks of high-dose (100 mg) melatonin supplementation positively affected the circadian components of the sleep–wake cycle among resistance-trained athletes [69]. There was an observed reduction in the nocturnal activity and a one-hour shift in the wrist temperature rhythm before bedtime, with a more extended nocturnal steady state and a smaller decrease during morning wake-up compared to the placebo group [69].

Similarly, the positive effect of melatonin intake on sleep quality was confirmed by Cheikh et al. in teenage athletes. A single 10 mg dose of melatonin consumption after exhaustive, late-evening training improved the total sleep time, sleep efficiency, stage 3 sleep, and REM phase compared to the placebo group. A reduction in sleep onset latency and the total time of nocturnal awakenings was found in the melatonin group. Moreover, melatonin supplementation improved the subjective sleep quality, muscle soreness, reaction time, speed, and performance during intermittent exercises [70].

Another aspect indicates that melatonin ingestion after sleep deprivation at night may enhance sports performance. Paryab et al. pointed out that 6 mg of melatonin administration 30 min before training improved the dynamic and statistical balance, reaction time, and anaerobic power among student-athletes who underwent sleep deprivation [71]. The effect of melatonin was also underlined by Cheikh et al., who examined its effect on muscle soreness, oxidative stress, and the inflammatory state among young athletes subjected to late-evening intense exercises. Researchers observed that ingestion of 10 mg of melatonin after an evening workout increased the peak power and mean power during the sprint test the following morning, reduced the fatigue index, and attenuated hematologic parameters, including the level of creatine kinase (CK), CRP, lactate dehydrogenase (LDH) and aspartate aminotransferase (ASAT) [72]. These observations support the role of melatonin in muscle damage and inflammation prevention. Leonardo-Mendonca et al. showed that four weeks of high-dose (100 mg) melatonin administration decreased the LDH and CK plasma concentration in resistance-trained athletes, yielding skeletal muscle protection [73]. Melatonin also acts as a high-capacity ROS scavenger within mitochondria, which promotes the expression of antioxidant enzymes: superoxide dismutase (SOD), glutathione peroxidase (GPx), glutathione reductase (GR), and catalase (CAT) via signal transduction through melatonin receptors. Supplementation by athletes during the pre-competition preparation period who consumed 5 mg of melatonin before sleep for 30 days caused a reduction in the IL-6 and CRP levels and increased the activity of GPx and SOD1 [74]. However, the research findings are not conclusive. The positive effect on recovery and physical and cognitive performance was not confirmed in elite handball players after a single 6 mg dose of melatonin intake [75]. Moreover, Ghattassi et al. observed a decrease in strength after a single dose (8 mg) treatment of melatonin among soccer players [76].

To summarize, melatonin acts as a high-capacity ROS scavenger within mitochondria, promoting the expression of antioxidant enzymes. However, its effects on recovery and performance may vary among individuals and sports disciplines.

#### 3.1.2. Probiotics

The human gastrointestinal tract hosts a diverse array of microorganisms, estimated to exceed 10^14^ in number [77]. The most abundant population is bacteria; thus, maintaining the proper balance of bacteria species and their biodiversity and richness is critical to host health [78]. The gut microbiota can indirectly influence sports performance by modulating the immune system, oxidative stress, metabolic processes, and nutrient availability. Additionally, intestinal microorganisms impact the protein turnover, glycogen storage, biogenesis, and mitochondrial function through various pathways. Bidirectional communication between intestinal microbes and skeletal muscles has been established, forming the gut–muscle axis [79].

In addition, the intestine microbiota may also affect skeletal muscle function through modulation of the nervous system. Recently, multiple studies have supported the hypothesis of the gut–brain axis and the bidirectional cross-talk between these two organs. This is mainly because of afferent and efferent neurons located in the lamina propria of the intestinal mucosa, which allows signal transmission. Thus, the CNS may influence intestine function; conversely, signals from the gut lumen may affect the mood, mental health, and brain function [80,81]. The gut microbiome plays a crucial role in this gut–brain cross-talk due to its ability to synthesize certain neurotransmitters. It has been proved that some *Lactobacillus* strains may produce γ-aminobutyric acid (GABA), similarly *Bacillus mycoides* and *Bacillus subtilis genus*—dopamine; *Bacillus cereus*, *Bacillus mycoides* and *Bacillus subtilis genus*—noradrenaline; as well as *Lactococcus lactis*, *Lactobacillus plantarum* and *Enterococcus thermophilous genus*—serotonin production [80,82,83]. Moreover, the gut microbiome may improve mood and brain function through neurotransmitter production and modulation of inflammatory state and, thus, reduce the activity of the HPA axis [84].

The mutual relationship between the gut microbiota and the HPA axis is a widely accepted fact [85]. However, the mechanisms underlying this interaction must be more clearly elucidated. The gut microbiota may activate the HPA axis through several mediators that cross the BBB, including microbial antigens, cytokines, and prostaglandins [86]. On the other hand, some studies indicate that the vagus nerve is one of the major signaling components between the gut microbiota and the brain [87], and the notion that the vagus nerve is widely involved in modulating the function of peripheral cells and tissues became popular nearly two decades ago [88]. Based on the influence of the gut microbiome on brain function and behavior, the term psychobiotic was coined [89,90,91].

In the study by Crumeyrolle et al., germ-free mice (GF) demonstrated a higher serum corticosterone (CORT) concentration as well as elevated hypothalamus corticotropin-released factor mRNA expression and a lower hippocampus dopaminergic turnover rate compared with specific pathogen-free mice [92]. Similarly, Bravo et al. have shown that the stress-induced CORT level decreased due to *Lactobacillus rhamnosus* supplementation. Moreover, they have observed positive alternation in central GABA receptor expression and improvement of depression- and anxiety-dependent behavior [93]. In addition, the elevated CORT concentration is often associated with a reduction in protein synthesis, exacerbation of gluconeogenesis, and induction of oxidative stress [94]. All of these may lead to muscle atrophy and weakness [95,96].

The gut microbiome composition can affect sleep quality. Approximately 95% of serotonin is synthesized in the intestine, where bacteria play a crucial role. Furthermore, it has been established that elevated serotonin levels may lead to fatigue and sleep disturbance [97]. It seems that a proper *Firmicutes*/*Bacteroidetes* ratio and bacterial biodiversity were described as the main components enhancing sleep quality [98]. Conversely, short-term sleep loss can induce adverse changes in the gut microbiome content, promoting the growth of the *Coriobacteriaceae* and *Erysipelotrichaceae* families, combined with a lower abundance of the *Tenericutes* family [99].

In the study by Harnett et al. involving elite male rugby players, there has been presented a relationship between probiotic supplementation, muscle soreness, and sleep quality. The authors indicated lower self-reported muscle soreness, as well as the perception of leg heaviness, in the group supplemented with ultra-biotic (containing *Lactobacillus*, *Bifidobacterium*, and *Streptococcus genera*) and SBFloractiv (Bioceuticals Australia AustL# 285024, containing *Saccharomyces boulardii*) compared to the placebo group. It should be emphasized that leg heaviness and muscle soreness increased when motivation, sleep quality, and quantity declined [100]. Similarly, Marotta et al. observed an improved mood state and sleep quality among healthy adults after 6 weeks of probiotic-mixture intake (containing *Lactobacillus* and *Bifidobacterium genera*) [101]. However, there is a limited number of studies that have explored the effects of probiotics on sleep quality within the athletic population. Moreover, existing research on probiotics in sports performance has indicated that while supplementation with certain bacterial strains can have positive health effects, it does not always translate into improved exercise capacity. For example, in a study conducted by Townsend et al., baseball players were supplemented with *Bacillus subtilis* (DE111), which led to a decrease in the TNF-α concentration. However, no effects were observed on the testosterone and cortisol concentration, strength or sports performance [102]. Similarly, a study conducted by Toohey et al. confirmed that *Bacillus subtilis* supplementation among volleyball players did not enhance sports performance [103].

There is some evidence suggesting that some bacteria strains may enhance sleep quality among inactive individuals. Takada et al. showed that 8 weeks of the *Lactobacillus* casein strain Shirota had a beneficial effect on sleep quality under psychological stress. In the group of healthy students preparing for an examination, probiotic intake had a significant positive effect on sleep quality, which manifested through the suppression of sleep lancet, maintenance of N3 sleep, or an increase in the delta power during the first sleep cycle. The results suggest that some bacteria species may enhance sleep quality during increasing stress [104]. Another study involving medical students showed an improvement in the depressive mood, anxiety, and sleep quality after 4 weeks of *Lactobacillus gasseri* CP2305 compared to the placebo [105]. The mechanism may be related to microbiotas’ ability to synthesize neuromolecules, including GABA, melatonin, and acetylcholine. Finally, a recent meta-analysis has underlined that some bacteria species were able to reduce depressive symptoms in both healthy adults and individuals with depression. Of interest here is the beneficial effect of a broad range of probiotic supplementation, which is not limited to clinically significant mood dysfunction but also may have a beneficial effect on the mental state in the population of healthy adults [106].

Overall, the dynamic interplay between the gut microbiota, the gut–brain axis, and the HPA axis holds significant implications for exercise, mental well-being, and sleep quality, warranting further exploration in both athletic and non-athletic populations.

#### 3.1.3. Vitamin D

In recent years, there has been an increased interest in the vitamin D status in athletes. A growing body of evidence has confirmed the presence of vitamin D receptors (VDRs) in nearly all human tissues [107]. Via binding to nuclear VDRs, active vitamin D (1,25(OH)D_3_; calcitriol) can regulate key metabolic processes influencing human health through autocrine and endocrine mechanisms [108]. It appears that the physiological role of vitamin D and its mechanism of action are multidirectional and complex, concerning the regulation of the nervous-, endocrine-, muscular-, immune- and cardiovascular systems, bone mineralization, regulation of gene expression, and hormone production [107]. Moreover, it is essential for energy metabolism, oxidative stress, maintenance, and improvement of physical fitness [109,110]. However, vitamin D deficiency is common and widespread among athletes, depending on the geographical location and type of sports discipline [111]. In addition, an adequate level of this hormone may be beneficial for athletes’ sports performance, primarily via favorable protein synthesis, increased skeletal muscle strength and health, decreased injury rates, and finally, enhanced CNS and brain health, and reduced stress response [112,113].

Vitamin D is difficult to obtain with the diet due to its low content in food. Therefore, it is mainly synthesized in the skin after sun exposure through the interaction of 7-dehydrocholesterol stored in the skin cells with ultraviolet B (UVB) radiation [114]. Vitamin D delivered from food, sun exposure, or supplements is biologically inert and must be doubled hydroxylated to become active by two enzymes: CYP2R1 in the liver and CYP27B1 in the kidneys. As a result, an active form of vitamin D, 1,25-dihydroxyvitamin D_3_, is created and transported with vitamin D-binding protein (BPD) in the bloodstream, reaching numerous targeted organs, including the skeletal muscles and nervous system cells [115].

The musculoskeletal benefits of vitamin D are widely known [114]. Some studies suggest that the vitamin D concentration in blood is closely associated with bone mineral density and high-intensity training is believed to compensate for vitamin D deficiency. However, scientific evidence is inconsistent in the athletic population [116,117]. Similarly, there is some evidence that the role of vitamin D is frequently observed in athletes’ stress fractures. A literature review indicates that vitamin D may enhance the soft callus formation phase during the recovery period of stress features. However, other researchers have not found any statistically significant effect on soft callus formation or the mineralization phase [111]. Active metabolites of vitamin D are also known as a modulator of skeletal muscle physiology through activating the gene expression responsible for myocyte growth and differentiation. Vitamin D exerts an effect on muscle function via VDR in both genomic (transcription and translation of targeted genes) and nongenomic (associated with the membrane, enhancing interaction between myosin and actin and thus increasing muscle sarcomere contraction force) pathways [118,119]. A low serum vitamin D concentration in athletes may interrupt their muscle strength and athletic sports performance [120] and delay recovery after orthopedic injury [119]. Some studies suggest that when an optimal vitamin D concentration is achieved, certain muscle performance parameters improve [121,122,123], and injury occurrence decreases [124]. On the other hand, some studies indicate no effect of vitamin D_3_ supplementation on sports performance. In the study conducted by Hewbutler et al., 12-week supplementation with 4000 IU vitamin D_3_ in basketball players did not enhance the peak power output [125]. It appears that optimal vitamin D intake may be beneficial for athletes by increasing the synthesis of muscle proteins, preventing muscle degradation, accelerating recovery after a workout, and enhancing sports performance [111]. The precise mechanisms that promote the storage and release of 25(OH)D_3_ and other vitamin D metabolites from these tissues are still unclear. However, recently, we showed that physical activity may play a role in this process. In the study by Dzik et al., a single bout of exercise in young subjects was sufficient to elevate the serum 25(OH)D_3_ concentration [126].

Moreover, vitamin D may indirectly affect the skeletal muscles and sports performance via modulation of the central and peripheral nervous systems. It has been proved that VDRs are also located in the brain, mainly in the hypothalamus, including the paraventricular nucleus [112], but also in the motor cortex, the region responsible for movement coordination, and the hippocampus [127]. In addition, vitamin D metabolites influence neurotrophic factors’ synthesis expression of synapse structural proteins and neuroactive molecules synthesis like GABA, dopamine, and serotonin, and they may affect nociceptors. It has been shown that vitamin D deficiency correlates with higher activity of nociceptors in deep muscle tissue, causing balance loss without any effect on muscle strength [127]. A study conducted by Seyedi et al. has indicated that 12 weeks of supplementation with vitamin D_3_ (2000 IU/day) in ADHD children elevated the serum dopamine with any effect on the brain-derived neurotrophic factor (BDNF) and serotonin concentrations [128]. However, it has been shown that vitamin D positively affects the nervous system by modulation of the kynurenine cycle pathway. Calcitriol is known to stimulate the expression of tryptophan hydroxylase-2 (TPH2) in brain cells, an enzyme responsible for serotonin biosynthesis [129]. Thus, vitamin D, via modulation of the neurotransmitter levels, is crucial for muscle coordination and avoiding central fatigue. A high proportion of serotonin and dopamine affects the exercise capacity due to its effect on tiredness and subjective perceptions of effort.

Some scientific evidence suggests that vitamin D may regulate the stress axis. Low vitamin D concentrations have been linked to dysregulation of the HPA axis and depression [130]. It has been shown that vitamin D can suppress cytotoxicity in hippocampal cells triggered by glucocorticoid-induced transcription [112]. Smolders et al. have indicated that corticotrophin-releasing hormones may be responsive to vitamin D due to positive staining for vitamin D 24-hydroxylase [131]. Furthermore, a study by Rolf et al. found a trend toward reducing the cortisol awakening response by vitamin D_3_ supplementation in female multiple sclerosis patients over 16 weeks. This finding suggests an appropriate outcome to investigate the suppression of the HPA axis by vitamin D_3_ [132]. Another study reported an improvement in depression severity with 8 weeks of vitamin D_3_ supplementation (50,000 IU/2 weeks) with mild/moderate depression compared to the control group [130]. In a group of forensic inpatients, it was noted that the positive effect of vitamin D on the stress response varied with seasonal changes in the vitamin D concentration, although no effect on the cortisol and serotonin levels was observed [133]. Finally, some data suggest that vitamin D could improve anxiety symptoms [134]. Taken together, the overactivation of the HPA axis hurts skeletal muscle metabolism and athletes’ mood and recovery processes.

It seems that an optimum level (≥50 ng/mL) of 25(OH)D_3_ may benefit sleep quality and thus improve regeneration processes, concentration, and athletes’ well-being. Recent interest in the role of this hormone indicates the neuronal expression of *VDR* and *CYP27B1* in certain brain areas, especially the hypothalamus [135]. The study conducted by Bertisch et al. reported that a decreased serum 25(OH)D_3_ level was associated with a shorter sleep time as measured by polysomnography [136]. Similarly, Hansen et al. found a positive correlation between 25(OH)D_3_ and sleep efficiency [133]. Interestingly, in particular studies, individuals with narcolepsy with cataplexy had lower 25(OH)D_3_ levels than healthy individuals, although Dauvilliers et al. did not observe any differences [137]. In the study by Rorie et al., supplementation with high-dose vitamin D_3_ (4000 IU/day) for 12 weeks significantly improved the subjective feeling of sleep quality compared to individuals who took low doses (600 IU/day) [138]. Some studies suggest that pro-inflammatory markers may be involved in sleep impairment. It was found that IL-1 and TNF-α exhibited inverse relationships with 25(OH)D_3_ [139,140]. Moreover, 25(OH)D_3_ inhibited TNF-α production following stimulation by lipopolysaccharides [141]. Thus, the associations between sleep disorders and blood vitamin D levels are still unclear. Therefore, further studies are required, especially in the athlete population.

#### 3.1.4. Polyphenol-Rich Diet

Polyphenols are organic compounds of plant origin found in fruits (plums, berries, grapes), vegetables (broccoli, lettuce), and beverages (green tea, red wine), among others (cocoa, turmeric) [142]. Depending on their structure and origin, polyphenols can be categorized into flavonoids (flavonols, flavanones, flavanols, flavones, isoflavones, and anthocyanins), phenolic acids, stilbenes, and lignans [143]. Through metabolic interactions with the intestinal microbiota, polyphenols, as bioactive phenolic metabolites, exhibit many health-related properties, including anti-inflammatory and antioxidant effects [144,145]. A 10-day supplementation of resveratrol (500 and 1000 mg/day) to young non-exercising men has been shown to reduce the neutrophil and lymphocyte counts and CK levels compared to the placebo group 24 h after plyometric exercise and exercise-induced muscle damage (EIMD). Moreover, it increases exercise performance, effectively reduces muscle pain, and improves physical adaptation [146]. Interestingly, diets rich in polyphenolic compounds also affect the composition of the microbiome itself. It has been shown that 2 weeks of blackcurrant extract powder (672 mg) treatment promotes an increase in the population size of *Bifidobacterium* and *Lactobacillus* in the gut microbiome of healthy voluntaries [147]. It influences the *Bacteroides*/*Firmicutes* balance and, at the same time, decreases the growth of pathogenic *Clostridium* species [148]. Polyphenol-induced variation in the gut microbiome translates into communication with the CNS via the vagus nerve, metabolites, and neurotransmission, among other effects. It increases the synthesis of GABA, BDNF, Trp, short-chain fatty acids (SCFAs), and serotonin [149]. Sadowska-Krępa et al. showed that a 6-week intake of green tea extract (245 mg) by male CrossFit athletes affects aerobic capacity and increases the serum BDNF levels [150].

Recent research has demonstrated the protective properties of polyphenols in relation to the BBB, which can become compromised under ROS generation, resulting in the loss of integrity and increased permeability [151]. Furthermore, it is emerging that polyphenols can selectively permeate the BBB and have been shown to attenuate neuroinflammatory processes by modulating the NF-κB pathway [152]. Additionally, by stimulating mitochondrial biogenesis, polyphenols improve cellular oxygen metabolism and ameliorate the expression of genes coding proteins that activate sirtuins and have cytoprotective activity [153,154]. A 15-day intake of pomegranate extract (225 mg punicalagins/day) by unprofessional cyclists has been shown to meaningfully affect performance during maximal efforts after long-term submaximal training and may help restore the strength in damaged muscle [155]. These properties suggest that polyphenol supplementation may enhance athletic performance and expedite post-workout muscle recovery through the gut–brain–muscle axis.

One of the main challenges of polyphenol supplementation seems to be their bioavailability and the individual variability of athletes. The absorption of polyphenols is determined by external factors (sun exposure, hydration, fruit ripeness), food processing (heat treatment, storage, homogenization), interaction with other substances, the health of the intestinal microbiota, and intestinal absorption [156]. However, metabolites formed as a result of bacterial degradation may locally modulate the composition of the microbiota, thereby indirectly impacting metabolism and bioavailability [157]. Recent studies have shown that polyphenols protect the blood–brain barrier and improve mitochondrial function, enhancing overall cellular metabolism. Despite these benefits, the challenges in terms of polyphenol supplementation include bioavailability and individual variability, influenced by many factors. Therefore, in this review, we also present substantial evidence suggesting that a polyphenol-rich diet may have no impact on sports performance and commonly measured blood markers.

#### 3.1.5. Beetroot

Beetroot is considered one of the most important vegetables that should be included in a balanced diet. It is rich in betaine, choline, polyphenols, flavonoids, anthocyanins, and phenolic acids, including *p*-coumaric, protocatechuic, ferulic, and vanillic acids, as well as *p*-hydroxybenzoic [158]. In addition, beetroot is rich in nitrates (NO_3_^−^), which serve as precursors of nitric oxide (NO). A diet high in nitrates may be crucial for athletes, as it has the potential to enhance tolerance to high-intensity exercise and reduce the oxygen cost during submaximal exercise [159]. The physiological benefits of NO include vasodilation and increased blood flow to muscle fibers, promoting oxygen delivery. In addition, by stimulating gene expression, NO affects mitochondrial biogenesis and efficiency [160]. Moreover, NO serves multiple functions, including the modulation of the release of acetylcholine, dopamine, GABA, serotonin, norepinephrine, and glutamate. NO is also known to stimulate the secretion of certain hormones, such as cortisol and testosterone. NO regulates adrenergic receptors associated with the autonomic nervous system by stimulating guanylyl cyclase and increasing the cyclic guanosine monophosphate (cGMP) levels. As a result, NO may exert a positive inotropic effect through this mechanism involving adrenalin and noradrenalin [161]. In addition, increasing its secretion regulates fatigue during prolonged activity, positively affects skeletal muscle metabolism and development, and modulates the energy balance and glucose uptake by skeletal muscle [162]. It has been confirmed that NO can modulate cortisol secretion by affecting HPA axis components such as the pituitary and adrenal cortex [163]. By activating the HPA axis associated with training, NO increases the availability of metabolic substrates, thereby improving performance [164]. Inorganic nitrate supplementation has been shown to improve short-term endurance training performance and the accompanying mental fatigue, in contrast to long-term performance [165]. Ingestion of NO_3_^−^ rich beetroot juice (BJ) by male students increased the peak power output and decreased the time to target during the performance of the Wingate test compared to the placebo group. In addition, a single BJ supply 3 h before training reduced the post-exercise ratings of perceived muscle exertion [166]. Furthermore, Clifford et al. showed that 72 h BJ intake (2 × 250 mL) accelerated the countermovement jump (CMJ) and reactive strength index (RI) after a repeated sprint test (RST) compared to the placebo group. While a time effect was observed on the serum CK levels and lipid hydroperoxides (LOOH) concentration, no group interactions were reported for CRP, CK, LOOH, protein oxidation markers, or ascorbyl free radical levels [167]. However, a study conducted by Kozlowska et al. on a group of elite fencers revealed that a diet supplemented with freeze-dried BJ at a dose of 26 g in 200 mL/day increased lipid peroxidation, and at the same time, improved the athletes’ performance. Another group demonstrated that 7-day consumption of BJ reduced the perceived muscle soreness (DOMS) after a simulated football game and enhanced performance during the recovery period in athletes [168]. BJ supplementation does not significantly affect between-group differences in the LDH, CK, and CRP concentration [169]. Recent research results are inconsistent regarding the positive effects of BJ on preparing athletes for improved athletic performance. Weekly supplementation of 3 × 70 mL/day improved athletes’ performance during the submaximal running test and increased time to exhaustion (TTE) during the cycling exhaustion test, while BJ did not improve performance during the 10 km skiing competition [170].

The inconsistency of the studies may be due to the degree of training of the study group. However, the most influential factor seems to be the supplement dose, the frequency of the daily intake, and the total supplementation period. Daily supplementation of beetroot juice twice a day at 150–250 mL (1–2 beetroot) seems to have the most significant effect on improving performance and lowering perceived DOMS. Nevertheless, it is imperative to acknowledge that the responses of individuals to beetroot treatment may differ. However, it should be remembered that consuming large amounts of NO-containing beets can produce *N*-nitroso compounds under metabolism, which belong to the class of potentially carcinogenic compounds [171]. Further research is needed to fully understand the potential side effects and contraindications associated with beetroot treatment, as current data on the subject are limited.

#### 3.1.6. Curcumin

The source of curcumin is the rhizomes of turmeric (*Curcuma longa*). Curcumin is well known for its pleiotropic effects, which positively influence immunomodulation, metabolic regulation, neuroprotection, and tissue protection. This is due to its antioxidant and anti-inflammatory properties [172]. Moreover, in subjects with a VO_2max_ of 65%, it was shown that 3 days of 500 mg curcumin supplementation improved the gastrointestinal barrier function and reduced the associated levels of the cytokine interleukin-1 receptor antagonist (IL-1RA) and the level of intestinal fatty acid-binding protein (I-FABP) during heat stress training [173]. Curcumin modulate reduces the production of prostaglandins and the pro-inflammatory cytokines interleukin 1β, 6, 8 (IL-1β, IL-6, IL-8), and TNF-α by modifying cyclooxygenase-2 (COX-2) signaling [174,175]. Mallard et al. showed that supplementation of healthy men with 450 mg of curcumin extract 30 min before resistance training decreases the LA and reduces the post-workout pain 48 and 72 h after training. These results have connected with modulating the levels of IL-6 upward and, consequently, IL-10. Interestingly, at the same time points, no differences were observed in the levels of CK, LDH, CRP, myoglobin, and TNF-α [176]. Curcumin supplementation appears to decrease lipid peroxidation and increase the antioxidant potential [177]. McFarlin et al. reported that 6 days (2 days before and 4 days after eccentric training) supplementation with 400 mg of curcumin reduced the CK levels by nearly 50%. The concentration of the pro-inflammatory cytokines IL-8 and TNF-α was reduced by about 20% after eccentric exercise compared to the placebo group. However, there was no difference in the concentrations of IL-6 and IL-10 and no meaningful change in the levels of muscle soreness between groups [178].

The study by Sciberras et al. focused on the supplementation of male recreational athletes with 500 mg of Meriva^®^ curcumin once every 3 days before exercise and 500 mg just before exercise. They found that curcumin supplementation had a lowering effect on the IL-6 and IL1-RA levels after endurance exercise compared to the control and placebo. In addition, participants experienced improvements in their subjective psychological stress scores [179]. Subsequent research conducted with 20 healthy males showed that 28 days of 1.5 g/day curcumin supplementation significantly reduced the CK levels compared to the placebo group. However, it did not significantly affect the total antioxidant capacity, markers of oxidative stress, and inflammation (MDA, TNF-α) [180]. The study by Salehi et al., showed that 80 women who participated in an 8-week course of curcumin treatment (500 mg/day) significantly improved their VO_2max_, CRP, LDH, and MDA levels. In contrast, the results for the anthropometric indicators presented no significant differences [181].

It was shown that weekly (twice a day) supplementation with curcumin extract (250 mg) and piperine (10 mg) in 16 trained male runners did not increase physical performance during exercise. In addition, it had no significant effect on the leukocyte count or muscle damage. However, it has been shown that supplementation with curcumin and piperine modulates IL-2, TNF-α, INF, IL-6, and IL-10 one hour after exercise [182].

The inconsistencies in the study results may stem from various factors, including the dose, duration of supplementation, study population, or type of training. The most significant factor appears to be the bioavailability of curcumin, which remains a contentious issue due to its poor absorption despite availability at high doses.

#### 3.1.7. Tart Cherry

The fruits of this plant are rich in anthocyanidins (cyanidin, peonidin), flavonols (isorhamnetin, kaempferol, quercetin), and catechins and epicatechins. In addition, the fruits of tart cherry (TC) are rich in melatonin, which may also have a beneficial effect on sleep and muscle regeneration [183]. It has been shown that the tart cherry compounds with the highest antioxidant activity exhibit a strong synergistic effect and modulate certain cellular pathways [184]. The latest research results suggest that a 10-day TC supplementation of 30 mL twice per day helps normalize the level of isometric muscle strength after exercise. In addition, increased mRNA expression of antioxidant enzymes such as superoxide dismutase 3 (SOD3), glutathione peroxidase 1 (GPx1), 3, 4, and 7 was observed in the treated group. However, there were no significant differences in the plasma levels of IL-6, TNF-α, CRP, and CK between the groups [185]. Other authors have demonstrated the efficacy of an 8-day TC intake of 30 mL twice per day on the markers of muscle recovery after prolonged intermittent sprint activity. Seventy-two hours after training, a reduction in muscle DOMS, faster recovery of performance indices (maximal voluntary isometric contraction [MVIC], CMJ, and agility), and a reduction in the IL-6 levels were demonstrated in the supplemented group. There were no significant changes in the LOOH and CK levels [186]. Another study evaluated the effects of 2 × 30 mL/day Montmorency cherry supplementation for eight days on a player’s recovery after a rugby league match. The researchers found no significant differences in the muscle DOMS, CMJ, distance traveled, or cytokine levels of IL-6, IL-8, and IL-10 compared to the placebo [187]. It has been shown that consuming 90 mL of TCJ concentrate daily for six days does not influence the perception of DOMS. Additionally, TCJ did not affect the post-workout levels of IL-6 and CRP in the bloodstream, nor did it reduce oxidative stress in water polo athletes [188].

On the other hand, Quinlan and Hill demonstrated the effect of 8 days of TC supplementation (30 mL; twice per day) in terms of recovery from intermittent exercise. They found that athletes treated with TC performed better in the CMJ, 20 m sprint, and MVIC, while there were no significant differences in the serum CK levels compared with the placebo [189]. These data suggest that even the short-term consumption of TC juice can improve sleep quality. This is confirmed by a study conducted by Chung et al. [190] on field hockey players who consumed 200 mL of TC juice containing 213 ± 41 μg/g anthocyanins 5 times over 48 h. The researchers showed that despite no changes in the melatonin and cortisol levels, the players improved sleep quality after intermittent training. Consumption of TC juice may be an important strategy for improving sleep quality and recovery during the sports season. Furthermore, using a blend of TC prolongs the sleep duration, increases sleep efficiency, and may regulate neurophysiological modulation of the sleep cycle by regulating the key metabolites involved in these processes. The importance of sleep in an athlete’s career appears crucial for performance, motivation, cognitive processes, and levels of perceived exertion and pain perception [191]. Although supplementation has not been shown to increase the melatonin levels, it appears that tart cherry consumption can modulate sennogenic cytokines such as IL-1B, IL-8, and TNF-1α [192] while lowering the IL-6 levels and reducing muscle pain [186]. Although the timing of the supplementation as well as the volume of fluid intake were consistent in the studies conducted, the composition of the TC juices differed significantly (concentration of polyphenols and anthocyanins).

The discrepancies in the presented results may be due to various factors, including the degree of training of the study group, the training procedure, the specific type and intensity of the exercise performed, and individual variability among participants. Other contributing factors could include the timing of the supplementation relative to the exercise, the form of the selected supplements used (e.g., standard extract vs. enhanced bioavailability formulations), diet, lifestyle factors, and genetic differences affecting metabolism. The studies using selected supplements (polyphenols, beetroot, curcumin, and tart cherry) are summarized in Table 1.

## 4. Muscle Recovery

### Remote Ischemic Conditioning—Non-Nutritional Muscle Regeneration

Remote ischemic conditioning (RIC) is a phenomenon in which short ischemia–reperfusion cycles protect against future prolonged ischemic events. Typically, RIC is applied with pneumatic cuffs in the proximal part of limbs in 3 to 5 cycles of 5 min ischemia separated by reperfusion [193]. Recently, RIC has been used to accelerate post-exercise recovery and reduce the effects of exercise-induced muscle damage (EIMD) [194]. EIMD causes the disturbance of calcium ion (Ca^2+^) homeostasis, the excessive response of the immune system, and oxidative stress, which results in pain and swelling of the muscles and decreased exercise performance [195]. Apart from the origin of skeletal muscle damage, similar pathophysiology appears in ischemic reperfusion injury (IR), which RIC can alleviate [196,197]. The similar pathophysiology of IR and EIMD was the primary motive for using RIC in accelerating post-exercise regeneration. Since Beaven and colleagues first used RIC in the context of post-exercise regeneration, the interest of researchers has increased, which has resulted in new studies assessing RIC’s effect on muscle regeneration [198] (Table 2). The accepted determinants of the occurrence of EIMD are a decrease in exercise performance, muscle pain, muscle swelling, and morphological changes, such as the presence of CK in the blood above the accepted norm [195]. The mentioned determinants were investigated in published studies (Table 2), and RIC was used in both pre-exercise and post-exercise variants (RIpreC, RIpostC) and various repetition forms (acute and chronic).

Although most of the studies reported a positive effect of RIC in mitigating EIMD [194,198,199,200,201,202], some of the studies did not show the advantage of RIC over the control and sham groups [203,204,205]. This inconsistency may result from the participants’ level of training. Similar observations were noticed in studies where RIC was employed to enhance the physical performance of athletes. It was found that individuals with lower levels of training showed more noticeable improvements, while highly trained individuals exhibited either less noticeable or no discernible response to RIC [175,183,184]. However, in the context of post-exercise regeneration, RIC has not been studied in a population of highly trained athletes. Moreover, the RIC volume regarding recovery is unclear and requires further study targeting the impact of the RIC volume on post-exercise regeneration in both types of subjects, highly trained and lower-trained athletes. The potential mechanism(s) responsible for the positive effect of RIC may be related to the modulation of the immune response, oxidative stress, Ca^2+^ metabolism, anti-apoptosis activity, autophagy, and improvement of blood vessel dilation and blood flow. The effect of RIC on the Ca^2+^ changes in the skeletal muscle is currently unknown; nevertheless, it is known that RIC affects the adenosine autocoid, which in turn inhibits the sodium–calcium exchanger (NCX) through the adenosine A1 receptor pathway, thereby preventing excessive Ca^2+^ entry into the cell [206,207,208]. RIC also induces NO and prostacyclin, which, together with adenosine, increase blood vessel dilatation and improve the blood flow [209,210,211]. Murphy and colleagues showed that RIC activates Nrf2, contributing to increased protection against excessive oxidative stress [212]. Hypothetically, NO-induced by RIC might cause s-nitrosylation of the cysteine residues of complex I of the ETC, thus inhibiting the generation of ROS [213]. RIC also has an immunosuppressive effect, as indicated by the suppression of pro-inflammatory genes in circulating leukocytes and the downregulation of NF-κB [214,215]. Recently, the downregulation of follistatin-like 1 protein (FSTL-1) and IL-6 was observed in marathon runners who had used RIC [216]. Muscle cell damage appears after a series of stressful exercises, leading to adaptation, such as muscle hypertrophy and energy metabolism improvement [217].

Nevertheless, excessive damage implies prolonged oxidative stress and inflammation, which might exacerbate necrotic and apoptotic cell death, hampering muscle adaptation [218,219]. Therefore, strategies for alleviating apoptosis might positively affect muscle regeneration in these cases. Animal studies reported an increased transcriptional response of B-cell lymphoma 2 (BCL2) in response to RIC in the skeletal muscle, suggesting RIC’s anti-apoptotic properties [220]. Murphy and colleagues further observed the pro-survival effects of RIC in humans by examining the genomic response in skeletal muscle biopsies, showing the increased expression of genes responsible for cell survival (HSP8, HSP40) and the downregulation of pro-apoptotic (caspases 8 and 9) gene expressions [212]. Autophagy is a process with pleiotropic effects on cell function. The primary role of autophagy is to “recycle” by degrading the cell content and recovering nutrients [221].

However, autophagy is a sophisticated process with a key effect on muscle regeneration. Autophagy is related to the activation of satellite cells (SCs), differentiation of monocyte cells to macrophages, and polarization from pro-inflammatory (M1) to anti-inflammatory (M2), which is directly related to muscle regeneration. Autophagy removes damaged muscle tissue and supports the resolution of inflammation [222]. In addition, targeted autophagy removes damaged mitochondria (mitophagy), which helps to preserve mitochondria function and alleviates oxidative stress [223]. The basal autophagy levels adapt as a result of training and are higher in trainees, thus contributing to better muscle regeneration and energy metabolism efficiency [224]. In the liver IR model, RIC has been shown to alleviate the damage by activating autophagy [225]. However, whether RIC activates autophagy in the EIMD model is uncertain because this has not yet been studied, so a well-planned study is required.

RIC is an interesting method of supporting muscle regeneration. However, some studies show the need for more effect, which might be related to the insufficiently understood molecular mechanisms responsible for RIC’s positive effect. In addition, RIC was mainly studied in active recreational men, and the RIC volume needs to be better documented. In the future, RIC research in muscle regeneration should include highly trained athletes, use several RIC volumes, and reveal the molecular mechanism(s) responsible for the observed changes.

In summary, while there is promising evidence suggesting the potential benefits of remote ischemic conditioning in muscle regeneration, further research is needed to establish the optimal protocols, understand individual variability, and address the potential limitations. As of now, the field is dynamic, and ongoing studies may provide additional insights into the application of RIC for muscle regeneration.

**Table 2 nutrients-16-01842-t002:** The research studies from 2012 to 2021 related to athletes using remote ischemic conditioning.

Research Paper (*n* = 10)	Subjects	Clinical Assess	Biochemical Assess	RIC, Ischemia/ Pressure/Limb	Muscle Recovery Accelerate
Daab et al., 2021 [169]	Semi-professional soccer players (*n* = 12)	Jumps, sprint, MVC, MS	CK, LDH, CRP	3 × 5 min/ 50 mmHg > SBP/ thigh	YES
Pizzo Junior et al., 2021 [226]	Young healthy (*n* = 80)	PRE, MS, TC, MVC	CK, LA	4 × 5/AOP and 40% > AOP/thigh	IN PROGRESS
Arriel et al., 2018 [199]	Trained cyclists (*n* = 28)	Wingate test, MS, PRS, RPE, HR	CK	2 × 5 min and 5 × 2 min/ 50 mmHg > SBP/ thigh	YES
Page et al., 2017 [201]	Healthy recreationally active (*n* = 16)	MIVC, jumps, MS, TC	CK	3 × 5 min/ 220 mmHg/ thigh	YES
Cerqueira et al., 2021 [204]	Young healthy (*n* = 30)	MVIT, MS, TC, ROM,	CK	4 × 5 min/AOP/ thigh	NO
Franz et al., 2018 [200]	Young healthy (*n* = 19)	TC, VAS, TMG,	CK	3 × 5 min/ 200 mmHg/ upper arm	YES
Patterson et al., 2021 [202]	Healthy recreationally active (*n* = 23)	TC, MS, CMJ, MVIC	CK	3 × 5 min/ 220 mmHg/ thigh	YES
Beaven et al., 2012 [198]	Healthy recreationally active (*n* = 14)	Jumps, sprint, leg press test	-	2 × 3 min/ 220 mmHg/ thigh	YES
Northey et al., 2016 [203]	Healthy well trained (*n* = 12)	MVC, jumps, MS, PRS	-	2 × 3 min/ 220 mmHg/ thigh	NO
Williams et al., 2018 [205]	Rugby player (*n* = 24)	Jumps, MS, PRS	CK, LA	2 × 3 min/ 60%AOP/thigh	NO

Used abbreviations: MS—muscle soreness; PRS—perceived recovery; RPE—perceived exertion; TC—thigh circumference; CK—creatine kinase, LDH—lactate dehydrogenase; CRP—serum C-reactive protein; LA—lactate acid; MVC—maximal voluntary contraction; VAS—visual analog scale; ROM—range of motion; HR—heart rate; MVIC—maximal voluntary isometric contraction; CMJ—countermovement jump; MVIT—maximal voluntary isometric torque; TMG—tensiomyography; AOP—arterial occlusion pressure; SBP—systolic blood pressure.

## 5. Conclusions

Overall, this review emphasizes the benefits of a well-managed lifestyle in enhancing athletic performance and muscle regeneration through intricate gut–brain–muscle connections. Daily factors such as prioritizing sleep hygiene, maintaining a balanced diet, selecting appropriate supplements, and utilizing post-workout support methods may significantly enhance bodily functions and elevate sports performance. We suggest that following the “SPARKS” rule will yield positive outcomes, whereas embracing the “SMOULDER” formula will have adverse effects on athletes’ sports performance.

Nutritional strategies, which include a variety of vitamins, antioxidants, and essential nutrients, not only improve sleep quality but also play a crucial role in fostering beneficial changes in the gut microbiome. Additionally, they modulate the protective and regenerative processes of the CNS and skeletal muscles. Regarding post-workout recovery and mitigating exercise-induced muscle damage, RIC emerges as a promising approach for regulating immune responses, minimizing oxidative stress, and enhancing anti-apoptotic effects. Together, these factors collectively act as “SPARKS”, expediting recovery and bolstering sports performance.

Moreover, the adoption of the “SPARKS” rule assumes a substantial role in the aging process, preserving the health of both the brain and the muscles.

In conclusion, it is important to highlight that the difficulty in providing clear guidelines for athletes is due to the variability of study designs. Each study uses different doses, exercise intensities, and protocols, involving a small number of subjects from various sports, sexes, and age groups, which makes the overall picture quite confusing. Therefore, strategies targeting all the aspects discussed in this review remain of interest to scientists, physicians, sports dietitians, coaches, and athletes as well.

## Figures and Tables

**Figure 1 nutrients-16-01842-f001:**
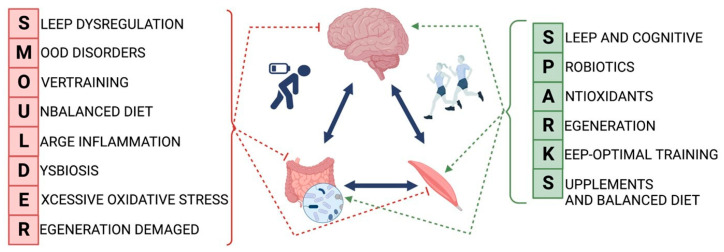
Impact of SPARKS and SMOULDER on sports performance.

**Table 1 nutrients-16-01842-t001:** The research studies from 2015 to 2023 related to athletes using supplements.

Research (*n* = 14)	Group	Supplementation	Dose	Clinical Assess	Biochemical Assess
Jodra et al., 2020 [166]	Young healthy (*n* = 15)	Beetroot juice	70 mL (6.4 mmol NO_3_^−^)	Profile of mood state, Wingate test, RPE scale	--
Clifford et al., 2016 [167]	Team-sports players (*n* = 20)	Beetroot juice	2 × 250 mL	RST, MIVC, CMJ, RI, pressure-pain threshold (PPT)	CRP, CK, LOOH, PC
Daab et al., 2021 [169]	Soccer players (*n* = 13)	Beetroot juice	2 × 150 mL	SJ, CMJ, maximal voluntary contraction (MVC), 20 m sprint DOMS	CK, LDH, CRP
Kozłowska et al., 2020 [168]	Elite fencers (*n* = 24)	Diet and freeze-dried beetroot juice	26 g in 200 mL/day	VO_2max_	CK, MDA, GPx1, GPx3, IL-6, LDH, AOPP
Huang et al., 2023 [170]	Winter triathletes (*n* = 80)	Beetroot juice	3 × 70 mL (6.5 mmol NO_3_^−^)/day	Submaximal treadmill run, intraday cycling exhaustion testing	LA
Wangdi et al., 2022 [185]	Young healthy (*n* = 10)	Montmorency cherry concentrate	2 × 30 ml	VAS, PPT, SLJ, MVC, IK^Max^, EC^Max^	SOD1, SOD3, GPx1, 3, 4, and 7, CAT, IL-6, TNF-α, CRP, CK
Bell et al., 2016 [186]	Semi-professional soccer players (*n* = 16)	Tart cherry juice	2 × 30 mL/day	DOMS MVIC, 20 m sprint, CMJ, and agility	IL-6 IL-1-β, IL-8, TNF-α, hsCRP, CK, LOOH
Morehen et al., 2021 [187]	Professional rugby players (*n* = 11)	Montmorency cherry	30 mL with 100 mL of water/twice per day	DOMS, subjective wellness, CMJ, DJ	IL-6, Il-8, IL-10
Quinlan and Hill, 2020 [189]	Team-sports players (*n* = 20)	Tart cherry juice	30 mL with 70 mL of water/twice a day	CMJ 20 m sprint MVIC	CK, CRP
Sciberras et al., 2015 [179]	Recreational athletes (*n* = 11)	Meriva^®^ curcumin	500 mg/day	HR/RPE	IL-6, IL1-RA, IL-10, CRP, CORT
McFarlin et al., 2016 [178]	Young healthy (*n* = 28)	Longvida^®^ curcumin	400 mg/day	DOMS, activities of daily living soreness	TNF-α, IL-6, IL-8, IL-10, CK
Salehi et al., 2021 [181]	Healthy women (*n* = 60)	Curcumin	500 mg/day	Body composition, QCT, FFQ	CRP, TAC, MDA, LDH, FRAP, TBARs
Mallard et al., 2020 [176]	Young healthy (*n* = 28)	Curcumin	250 mL (500 mg HydroCurc + 500 mg maltodextrin)	VAS, TC,	IL-6, IL-10, CRP, TNF-α, LDH, myoglobin, LA

Used abbreviations: RST—repeated sprint test; MVIC—maximal voluntary isometric contraction; CMJ—countermovement jump; PPT—pressure–pain threshold; RI—reactive strength index; SJ—squat jump; MVC—maximal voluntary contraction; DOMS—delayed onset muscle soreness; VAS—visual analog scale; SLJ—single leg jumps; IK^Max^—maximal isokinetic knee extension and flexion; EC^Max^—maximal eccentric contraction; DJ—drop jump; HR—heart rate; RPE—perceived exertion; QCT—Queen College step test; FFQ—Food Frequency Questionnaire; TC—thigh circumference; CRP—serum C-reactive protein; CK—creatine kinase, LOOH—lipid hydroperoxides; PC—protein carbonyl; LDH—lactate dehydrogenase; MDA—malondialdehyde; LA—lactate acid; GPx—glutathione peroxidase; AOPP—advanced oxidation protein products; SOD—superoxide dismutase; CAT—catalase; IL—interleukins; TNF-α—tumor necrosis factor; FRAP—ferric reducing/antioxidant power; TBARs—thiobarbituric acid-reactive substances.

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
