# Peer review of "Current Aspects of Selected Factors to Modulate Brain Health and Sports Performance in Athletes"

_nutrients, 2024, doi:10.3390/nu16121842_

Round 1

Reviewer 1 Report

Comments and Suggestions for Authors

Summary:  The authors constructed a current review of the positive and negative factors that impact brain health in regards to athletes, but also pointed out how these factors can impact others groups as well, including the growing aging population.

General comments/suggestions:

-add citations for line 35, and lines 74-76

-awkward wording in lines 77-79, consider rephrasing

-lines 99-101, consider using whole number OR a single decimal place through the paper for consistency.  Also, these are examples of stress on the general population, what what about examples of stress on athletes in particular?

-line 130 add citations, and what constitutes "adequate"?

-line 198 add citations, and when "sleep loss" is discussed, is this from an adequate amount to inadequate, or just reduced?

-has an athlete's microbiota ever been sufficiently characterized?

-do athlete's have a ideal gut composition for their respective sports?

Author Response

Reviewer #1

The authors of the manuscript would like to thank for all the comments and hints concerning improvements of this paper. We followed them to revise the manuscript or tried to explain any ambiguities in the point-by-point response.

Summary:  The authors constructed a current review of the positive and negative factors that impact brain health in regards to athletes, but also pointed out how these factors can impact others groups as well, including the growing aging population.

General comments/suggestions:

 -add citations for line 35, and lines 74-76

-awkward wording in lines 77-79, consider rephrasing

-lines 99-101, consider using whole number OR a single decimal place through the paper for consistency.  Also, these are examples of stress on the general population, what what about examples of stress on athletes in particular?

-line 130 add citations, and what constitutes "adequate"?

-line 198 add citations, and when "sleep loss" is discussed, is this from an adequate amount to inadequate, or just reduced?

Thank you for your comments and suggestions. We added citations, rewrote the sentence,  corrected the whole number in the manuscript, and added the quantity of sleep. Please review the manuscript with the tracked changes.

It is commonly assumed that due to a neurohormonal reaction to a stress stimulus, adrenal hormones are released into the general circulation: catecholamines, adrenaline, and noradrenaline from the adrenal medulla and glucocorticoids (GCs)  from the adrenal cortex [11].  This viewpoint has been accepted since it was recognized that understanding the systemic effects of these hormones requires considering their role in the body's adaptive response to environmental stress.

“Inadequate amount and quality of sleep may also cause a decline in mental functions and well-being [47]. Therefore, increasing sleep duration is associated with benefits in alertness, psychomotor functions, executive function performance, and motivation level [47]. To sum up, the data mentioned above strongly support the importance of sleep amount (seven or more hours) and quality is a critical factor promoting optimal regeneration processes, benefiting both the central nervous system and skeletal muscle tissue health”.

-has an athlete's microbiota ever been sufficiently characterized?

Thank you for this question. We hope that our answer will be satisfactory.

It is established that the microbiota of athletes differs from that of non-athletes in terms of biodiversity, species richness, and microbiota composition. Certain similarities in the composition of the gut microbiota can be observed among athletes, which highlights the interplay between physical activity and gut microbiota. However, research on athletes' microbiota is still evolving. Thus, it cannot be asserted that the composition of the gut microbiota in athletes has been sufficiently characterized.

Agnieszka Mika 1, Will Van Treuren 2, Antonio González 3, Jonathan J Herrera 1, Rob Knight 4, Monika Fleshner 1

Exercise is More Effective at Altering Gut Microbial Composition and Producing Stable

Changes in Lean Mass in Juvenile versus Adult Male F344 Rats. DOI: 10.1371/journal.pone.0125889

Wiley Barton 1 2 3, Nicholas C Penney 4 5, Owen Cronin 1 3, Isabel Garcia-Perez 4, Michael G Molloy 1 3, Elaine Holmes 4, Fergus Shanahan 1 3, Paul D Cotter 1 2, Orla O'Sullivan 1 2

The microbiome of professional athletes differs from that of more sedentary subjects in composition and particularly at the functional metabolic level.

DOI: 10.1136/gutjnl-2016-313627

-do athlete's have a ideal gut composition for their respective sports?

Thank you for this question. We hope that our answer will be satisfactory.

In light of current knowledge, it is impossible to determine whether there is an ideal gut microbiota composition tailored to the sports practiced. Available scientific evidence suggests that training can influence quantitative and qualitative changes in the composition of gut microbiota. For example, certain bacteria that metabolize lactate, such as Veillonella atypica, have been found in higher abundance in marathon runners and are thought to potentially improve endurance by converting lactate into propionate. However, the gut microbiota is influenced by numerous factors beyond physical activity, including diet, medication, and genetics. Therefore, it is crucial to consider gut microbiota holistically rather than focusing solely on individual bacteria. Consequently, probiotic supplementation aims to establish conditions conducive to maintaining an optimal gut microbiota composition, which is highly individual.

Jonathan Scheiman # 1 2 3, Jacob M Luber # 4 5 6 7 8, Theodore A Chavkin # 4 5 7, Tara MacDonald 9 10, Angela Tung 1 2, Loc-Duyen Pham 4 5, Marsha C Wibowo 4 5 7, Renee C Wurth 3 11, Sukanya Punthambaker 1 2, Braden T Tierney 4 5 6 7, Zhen Yang 4 5 12, Mohammad W Hattab 2, Julian Avila-Pacheco 8, Clary B Clish 8, Sarah Lessard 9 10, George M Church 13 14, Aleksandar D Kostic 15 16 17

Meta-omics analysis of elite athletes identifies a performance-enhancing microbe that functions via lactate metabolism.

DOI: 10.1038/s41591-019-0485-4

Nobuhiko Akazawa 1 2, Mariko Nakamura 3, Nobuhiko Eda 1 4, Haruka Murakami 5, Takashi Nakagata 5 6, Hinako Nanri 5 6, Jonguk Park 7, Koji Hosomi 8, Kenji Mizuguchi 7 9, Jun Kunisawa 8, Motohiko Miyachi 2 5, Masako Hoshikawa 1

Gut microbiota alternation with training periodization and physical fitness in Japanese elite athletes.

DOI: 10.3389/fspor.2023.1219345

Esther Gil-Hernández 1, Cristofer Ruiz-González 1, Miguel Rodriguez-Arrastia 2, Carmen Ropero-Padilla 2, Lola Rueda-Ruzafa 2, Nuria Sánchez-Labraca 2, Pablo Roman 2 3 4

Effect of gut microbiota modulation on sleep: a systematic review and meta-analysis of clinical trials.

DOI: 10.1093/nutrit/nuad027

Reviewer 2 Report

Comments and Suggestions for Authors

interesting and useful article

Author Response

Reviewer # 2

The authors of the manuscript would like to thank for all the comments and hints concerning improvements of this paper. We followed them to revise the manuscript or tried to explain any ambiguities in the point-by-point response.

This article on post-exercise recovery for athletes provides a comprehensive review of various strategies and their physiological mechanisms to optimize adaptation and performance. Here is an overview and critical evaluation of its key points:

Strengths:

  1. Comprehensive Scope:The article thoroughly covers multiple aspects of recovery, including nutritional strategies, brain modulation, muscle recovery, and the gut-brain-muscle axis.
  2. Scientific Rigor:It references a wide range of scientific studies, ensuring a solid evidence-based approach.
  3. Detail-Oriented:The explanations of physiological processes and mechanisms are detailed and informative, providing a deep understanding of how each strategy works.
  4. Practical Applications:The discussion on the practical applications of various recovery strategies for athletes and individuals with different activity levels is particularly useful.

Areas for Improvement:

  1. Structure and Flow:

- The structure could benefit from clearer section headings and subheadings to improve readability. For instance, distinct sections for each type of recovery strategy (e.g., nutritional strategies, brain modulation, muscle recovery) would enhance clarity.

Thank you for your suggestion.

We have added these sections.

- Some paragraphs are overly lengthy and could be broken down for better readability and comprehension.

Thank you for your suggestion. The authors of this review tried to shorten the text in the manuscript wherever possible.

  1. Consistency and Redundancy:

- There is some redundancy, particularly in discussing the effects of certain nutrients and supplements like melatonin and probiotics. Streamlining these sections could reduce repetition.

- Ensure consistency in terminology and units (e.g., using either "mg" or "milligrams" uniformly).

Thank you for your comment. We have reduced the text and corrected units using “mg”.

  1. Depth of Analysis:

- While the article provides extensive information on various recovery strategies, it sometimes lacks a critical analysis of conflicting studies or limitations of current research. A balanced discussion including potential drawbacks or gaps in the evidence would strengthen the review.

- The inclusion of a table summarizing key findings and recommendations for each recovery strategy would be beneficial.

Thank you for these recommendations. We have incorporated some sentences into the manuscript.

Moreover, Ghattassi et al. observed a decrease in strength after a single dose (8 mg) treatment of melatonin among soccer players [76].

Therefore, in this review, we also present substantial evidence suggesting that a polyphenol-rich diet may have no impact on sports performance and commonly measured blood markers.

It has been shown that consuming 90 ml of TCJ concentrate daily for six days does not influence the perception of DOMS. Additionally, TCJ did not affect reducing post-workout levels of IL-6 and CRP in the bloodstream, nor did it reduce oxidative stress in water polo athletes [189].

We agree with you, however, it is very difficult to find recommendations for the described strategies, for example:

  1. It is established that the microbiota of athletes differs from that of non-athletes in terms of biodiversity, species richness, and microbiota composition. Certain similarities in the composition of the gut microbiota can be observed among athletes, which highlights the interplay between physical activity and gut microbiota. However, research on athletes' microbiota is still evolving. Thus, it cannot be asserted that the composition of the gut microbiota in athletes has been sufficiently characterized.

Agnieszka Mika 1, Will Van Treuren 2, Antonio González 3, Jonathan J Herrera 1, Rob Knight 4, Monika Fleshner 1

Exercise is More Effective at Altering Gut Microbial Composition and Producing Stable

Changes in Lean Mass in Juvenile versus Adult Male F344 Rats. DOI: 10.1371/journal.pone.0125889

Wiley Barton 1 2 3, Nicholas C Penney 4 5, Owen Cronin 1 3, Isabel Garcia-Perez 4, Michael G Molloy 1 3, Elaine Holmes 4, Fergus Shanahan 1 3, Paul D Cotter 1 2, Orla O'Sullivan 1 2

The microbiome of professional athletes differs from that of more sedentary subjects in composition and particularly at the functional metabolic level.

DOI: 10.1136/gutjnl-2016-313627

  1. In light of current knowledge, it is impossible to determine whether there is an ideal gut microbiota composition tailored to the sports practiced. Available scientific evidence suggests that training can influence quantitative and qualitative changes in the composition of gut microbiota. For example, certain bacteria that metabolize lactate, such as Veillonella atypica, have been found in higher abundance in marathon runners and are thought to potentially improve endurance by converting lactate into propionate. However, the gut microbiota is influenced by numerous factors beyond physical activity, including diet, medication, and genetics. Therefore, it is crucial to consider gut microbiota holistically rather than focusing solely on individual bacteria. Consequently, probiotic supplementation aims to establish conditions conducive to maintaining an optimal gut microbiota composition, which is highly individual.

Jonathan Scheiman # 1 2 3, Jacob M Luber # 4 5 6 7 8, Theodore A Chavkin # 4 5 7, Tara MacDonald 9 10, Angela Tung 1 2, Loc-Duyen Pham 4 5, Marsha C Wibowo 4 5 7, Renee C Wurth 3 11, Sukanya Punthambaker 1 2, Braden T Tierney 4 5 6 7, Zhen Yang 4 5 12, Mohammad W Hattab 2, Julian Avila-Pacheco 8, Clary B Clish 8, Sarah Lessard 9 10, George M Church 13 14, Aleksandar D Kostic 15 16 17

Meta-omics analysis of elite athletes identifies a performance-enhancing microbe that functions via lactate metabolism.

DOI: 10.1038/s41591-019-0485-4

Nobuhiko Akazawa 1 2, Mariko Nakamura 3, Nobuhiko Eda 1 4, Haruka Murakami 5, Takashi Nakagata 5 6, Hinako Nanri 5 6, Jonguk Park 7, Koji Hosomi 8, Kenji Mizuguchi 7 9, Jun Kunisawa 8, Motohiko Miyachi 2 5, Masako Hoshikawa 1

Gut microbiota alternation with training periodization and physical fitness in Japanese elite athletes.

DOI: 10.3389/fspor.2023.1219345

Esther Gil-Hernández 1, Cristofer Ruiz-González 1, Miguel Rodriguez-Arrastia 2, Carmen Ropero-Padilla 2, Lola Rueda-Ruzafa 2, Nuria Sánchez-Labraca 2, Pablo Roman 2 3 4

Effect of gut microbiota modulation on sleep: a systematic review and meta-analysis of clinical trials.

DOI: 10.1093/nutrit/nuad027

Moreover, as suggested by Reviwer#3 we have added in the Conclusion section these sentences:

In conclusion, it is important to highlight that the difficulty in providing clear guidelines for athletes is due to the variability in study designs. Each study uses different doses, exercise intensities, and protocols, involving a small number of subjects from various sports, sexes, and age groups, which makes the overall picture quite confusing. Therefore, strategies targeting all aspects discussed in this review remain of interest to scientists, physicians, sports dietitians, coaches, and athletes as well.

The text of this review is already substantial, and we believe that adding additional tables with citations already described would only lengthen the manuscript further. We have only used tables specifically to present test results where discrepancies exist.

  1. Clarity in Technical Details:

- Certain technical details, such as biochemical pathways and molecular interactions, might be too complex for some readers. Simplifying these explanations or providing a glossary could help.

  1. Figures and Visual Aids:

- The article could benefit from more figures, diagrams, and visual aids to illustrate complex concepts, especially the physiological mechanisms described.

- The existing figure ("Figure 1. Impact of SPARKS and SMOULDER on sports performance") is not clearly explained or referenced in the text. Providing a detailed explanation and ensuring it aligns with the content would be helpful.

Thank you for this comment. We have added more details related to Figure 1. We apologize, but we are unable to add additional graphs. We believe that the presented data and our interpretation from a physiological standpoint are quite clear.

Athletes are exposed to several negative external and internal factors that may disrupt sports performance and recovery processes. A cascade of negative events associated with chronic stress can initiate overtraining and sleep disorders, resulting in mood changes. Poorly balanced nutrition may lead to dietary neglect, causing dysbiosis with increased inflammatory and oxidative stress that affects many tissues and organs, directly increasing the risk of injury or infection (SMOULDER). To prevent the negative effects of chronic stress, elements described by the acronym SPARKS should be routinely implemented as preventive measures. These measures encompass both psychological-cognitive aspects (quantity and quality of sleep and post-workout recovery) and ensuring proper bodily functions, including a balanced diet rich in probiotics, antioxidants, and supplements to improve the physiological function of not only athletes.

Specific Feedback:

- Introduction (Lines 1-45): This section effectively sets the stage for the review. However, a clearer outline of what each section will cover would improve reader guidance.

- Stress and Recovery (Lines 46-121): The discussion on stress, its physiological impacts, and the role of recovery strategies is well-articulated. However, integrating more specific examples or case studies could enhance practical relevance.

- Sleep (Lines 122-206):This section is detailed and informative. It could be improved by summarizing key points at the end, such as optimal sleep duration and quality metrics for athletes.

- Nutritional Strategies and Supplements (Lines 207-506):This extensive section covers several important supplements. Summarizing the effects of each supplement in a table format would aid comprehension.

- Probiotics, Vitamin D, Polyphenols, Beetroot, Curcumin, Tart Cherry (Lines 315-699):The detailed explanations of each supplement's impact on recovery and performance are excellent. However, critical analysis of conflicting studies or potential side effects would provide a more balanced view.

Thank you for these comments and suggestions.

We tried to introduce as much as we could to the whole body text. Please review the manuscript with the tracked changes.

We agree with you, and we tried to cut off or change the text as well as give some explanation for the described strategies, for example:

Beetroot

Line 595-601

“Although BJ supplementation does not significantly affect between-group differences in LDH, CK, and CRP concentration (Daab et al. 2021). Recent research results are inconsistent regarding the positive effects of BJ on preparing athletes for improved athletic performance. Weekly supplementation of 3 x 70 ml/day improved athletes' performance during the submaximal running test and increased time to exhaustion (TTE) during the cycling exhaustion test, while BJ did not improve performance during the 10 km skiing competition (Huang et al. 2023).”

Curcumin

Line 641-651

“It was shown that weekly (twice a day) supplementation with curcumin extract (250 mg) and piperine (10 mg) on 16 trained male runners did not increase physical performance in the exercise. In addition, it had no significant effect on leukocyte count or muscle damage. However, it has been shown that supplementation with curcumin and piperine modulates IL-2, TNF-α, INF, IL-6, and IL-10 one hour after exercise (Miranda-Castro et al. 2022).”

Polyphenols-rich diet

“Despite the numerous potential benefits of a polyphenol-rich diet, substantial evidence suggests it may have no impact on (10.1371/journal.pone.0072215; 10.3390/nu6010050; 10.1123/ijsnem.22.6.486.)”

Tart cherry

“It has been demonstrated that daily consumption of 90 ml of TCJ concentrate for six days does not affect the perception of delayed onset muscle soreness (DOMS). Furthermore, it has been shown that the post-workout attenuation of interleukin-6 (IL-6) and C-reactive protein (CRP) levels in the circulation, as well as the attenuation of oxidative stress in water polo training athletes, is not affected by TCJ”.

Similar outcomes were observed in the absence of alterations in inflammatory markers, as reported by Bowtell et al. and Beals et al.”

Rachel McCormick 1, Peter Peeling 1, Martyn Binnie 2, Brian Dawson 3, Marc Sim 4

Effect of tart cherry juice on recovery and next day performance in well-trained Water Polo players  

DOI: 10.1186/s12970-016-0151-x

Bowtell JL, Sumners DP, Dyer A, Fox P, Mileva KN. Montmorency cherry juice reduces muscle damage caused by intensive strength exercise. Med Sci Sports Exerc. 2011;43(8):1544–1551.

Beals, K.; Allison, K.F.; Darnell, M.; Lovalekar, M.; Baker, R.; Nieman, D.C.; Vodovotz, Y.; Lephart, S.M. The effects of a tart cherry beverage on reducing exercise-induced muscle soreness. Isokinet. Exerc. Sci. 201725, 53–63

DOI:10.3233/IES-160645

Conclusion:

Overall, the article provides a valuable resource for understanding post-exercise recovery in athletes. By enhancing the structure, reducing redundancy, and including more critical analysis and visual aids, the article could become an even more effective and reader-friendly resource.

Would you like me to help restructure specific sections or provide additional visual aids and summaries to improve the readability and impact of the article?

We hope that the answers and corrections we have added to the manuscript are satisfactory.

Reviewer 3 Report

Comments and Suggestions for Authors

The review is well done.

Only 3 points

1. I asked the authors to introduce the sub-chapter epidemiology of stress before stress prolegomena.

2. In Table 2, enter the number of subjects as in Table 1

3. In the conclusions, the authors must underline that it is tough to give guidelines because all the studies use different doses, different protocols, and a very low number of subjects with other sports, different sexes, and different ages, so everything is still very confusing.

This point shoud be clearly highligheted 

Comments on the Quality of English Language

Minor editing of English language required

Author Response

Reviewer #3

The authors of the manuscript would like to thank for all the comments and hints concerning improvements of this paper. We followed them to revise the manuscript or tried to explain any ambiguities in the point-by-point response.

  1. I asked the authors to introduce the sub-chapter epidemiology of stress before stress prolegomena.

Thank you for this suggestion.

We have changed this.

2. In Table 2, enter the number of subjects as in Table 1

Thank you for this comment.

We have added the number of participants in Table 2.

3. In the conclusions, the authors must underline that it is tough to give guidelines because all the studies use different doses, different protocols, and a very low number of subjects with other sports, different sexes, and different ages, so everything is still very confusing.

This point shoud be clearly highligheted 

Thank you for this comment.

We have incorporated sentences as follows:

In conclusion, it is important to highlight that the difficulty in providing clear guidelines for athletes is due to the variability in study designs. Each study uses different doses, exercise intensities, and protocols, involving a small number of subjects from various sports, sexes, and age groups, which makes the overall picture quite confusing. Therefore, strategies targeting all aspects discussed in this review remain of interest to scientists, physicians, sports dietitians, coaches, and athletes as well.
